# Family support after a family member's suicide: A qualitative exploration

Franziska Marek*, Nathalie Oexle

Department of Psychiatry and Psychotherapy II, Ulm University, Ulm, Guenzburg, Germany

* franziska.marek@uni-ulm.de

## Abstract

Suicide bereavement presents unique challenges that shape how grief is experienced and supported within relational networks. Although family members often serve as a primary source of support, the interpersonal dynamics underlying such support remain understudied. This study explores how individuals bereaved by a family member's suicide experience social support within the shared grief context of the family. Fifteen adults participated in semi-structured interviews conducted as part of a mixed-methods project in Germany, examining experiences and determinants of social support following suicide loss. Data were analyzed using qualitative content analysis, combining inductive and deductive coding. Two overarching themes were identified: (1) *Contextual factors of family support*, including grief reactions and coping patterns, shifts in family dynamics, and the distribution of support roles; and (2) *Characteristics of family support*, encompassing both supportive experiences—such as emotional closeness, open communication, and shared sense-making—and insufficient support—such as marginalization of grief and emotional neglect, often linked to pre-existing family strain. The availability and quality of support were influenced by protective buffering, relational withdrawal, discomfort surrounding suicide disclosure, and the reconfiguration of relationships after the loss. Extended family members played a significant role in assisting with childcare and relieving emotional burdens on grieving parents, although their support varied depending on attitudes toward suicide and levels of proactive engagement. Family support in suicide bereavement emerged as a dynamic process shaped by personal, relational, and sociocultural influences. These findings highlight the need for multilevel, integrative theoretical frameworks to capture the complexity of family support after suicide. This study underscores the value of systemic postvention strategies that foster emotional expression, shared sense-making, and stigma reduction, while recognizing that family may not be an accessible or supportive resource for all bereaved individuals.

**Data availability statement:** The datasets generated and analyzed during the current study are not publicly available due to ethical restrictions aimed at protecting participant confidentiality. Although all interview transcripts have been pseudonymized, they contain contextual details that, in combination, could potentially allow for participant identification. In accordance with the terms approved by the Ethics Commission at Ulm University (application number: 374/18), participants consented to the use of pseudonymized interview data for scientific analysis and the publication of selected, anonymized quotations in scholarly outputs, but did not consent to full public disclosure of the transcripts. A minimal dataset supporting the study's findings is provided inThe datasets generated and analyzed during the current study are not publicly available due to ethical restrictions aimed at protecting participant confidentiality. In accordance with the terms approved by the Ethics Commission at Ulm University (application number: 374/18), participants consented to the use of pseudonymized interview data for scientific analysis and the publication of selected anonymized quotations in scholarly outputs but did not consent to full public disclosure of the transcripts. A minimal dataset supporting the study's findings is provided in S2 Table (found in the Supporting information), which contains additional anonymized quotations and full versions of quotations cited in the manuscript. Access to the complete pseudonymized transcripts may be considered for qualified researchers upon reasonable request. Formal inquiries regarding data access and restrictions may be directed to the Research Secretariat of the Section Public Mental Health, Department of Psychiatry and Psychotherapy II, Ulm University (E-mail: sekretariat-psyII@uni-ulm.de). The corresponding author may be contacted as a secondary point of information." Author response in their Cover Letter: "In response to the journal's data policy requirements, we have revised the 'Data availability statement' in the manuscript to meet all PLOS ONE criteria. The updated statement specifies the reason for the restriction, names the responsible institution, and provides a non-author, institutional contact email for data access requests, in addition to the corresponding author as a secondary contact.which contains additional anonymized quotations and full versions of quotations cited in the manuscript. Access to the complete pseudonymized transcripts may be granted to qualified researchers upon reasonable request,

## Introduction

Grief is often framed as a personal experience, yet it is profoundly shaped by relational and social contexts, including the circumstances of the death itself [1]. Suicide bereavement is associated with distinct emotional and psychosocial challenges, including guilt, social stigma, and strained interpersonal dynamics, that can complicate coping and limit access to support [2,3]. Each year, over 700,000 people die by suicide worldwide [4], with each death affecting a wide network of individuals [5]. As Cerel and colleagues [6] have emphasized, suicide loss occurs along a continuum of exposure and impact—from individuals who knew the deceased, to those who are emotionally affected, to those who are bereaved following the loss of a close attachment and experience profound psychological and relational consequences. Suicide-bereaved individuals, commonly referred to as suicide loss survivors (SLS), are at heightened risk for a range of adverse health outcomes, including depression [7,8], anxiety disorders and posttraumatic stress [9], suicidal ideation [7], suicide attempts [10], and complicated grief [11]. While these risks are well documented, scholars have cautioned that a purely diagnostic lens may obscure the broader psychosocial and cultural dimensions of bereavement [1]. As Breen and O'Connor [1] argue, such a perspective risks individualizing grief responses that are, in fact, shaped by socio-cultural influences, such as stigma and social marginalization. From this standpoint, suicide bereavement represents not only a mental health issue, but also a significant public health concern, with implications for social integration and family relationships. Yet the question of how families support each other in the aftermath of a family member's suicide—amid the complexities and challenges such loss entails—remains under-researched. Gaining deeper insight into these interpersonal dynamics is essential for developing postvention strategies that foster supportive family interactions and help mitigate health risks among SLS.

### Grief as a relational process

Grief is not only an individual reaction to loss but a relational process embedded within social systems—most notably, the family [12]. The death of a family member affects the familial system, altering roles, responsibilities, communication patterns, and relationships. Gilbert [13], adopting a social constructionist perspective on grief, conceptualizes the family as a central "arena" of bereavement. She argues that although family members may share "the same loss," they often do not experience "the same grief" due to differences in their relationships with the deceased, emotional expression, and coping strategies. This phenomenon, which she terms *differential grief*, can contribute to misunderstanding and emotional disconnection among family members.

Building on this perspective, Hayslip and Page [14], drawing from family psychology and systems theory, adopt a family systems framework that emphasizes the concept of *multileveled embeddedness*—the idea that grief unfolds within and across interrelated systems, including the family unit, broader social networks, and cultural or institutional contexts. From this view, families are dynamic systems whose

subject to review and approval by the responsible institutional bodies of Ulm University, in consultation with the Data Protection Officer. Formal inquiries regarding data access should be directed to the Research Secretariat of the Section Public Mental Health, Department of Psychiatry and Psychotherapy II, Ulm University (email: sekretariat-pmh-psy2@uni-ulm.de), which will coordinate the review process and ensure compliance with applicable data protection regulations.

**Funding:** NO (Principal Investigator of the DE-LOSS project) received funding for this research from the German Research Foundation (DFG – Deutsche Forschungsgemeinschaft), grant number 452241355. Funder website: https://www.dfg.de. The funder had no role in study design, data collection and analysis, decision to publish, or preparation of the manuscript.

**Competing interests:** The authors have declared that no competing interests exist.

communication patterns, roles and responsibilities (e.g., caregiving for relatives, raising children), intergenerational composition, and the specific circumstances of the death shape the grieving process. This framework underscores the crucial role of the family in how grief is expressed, processed, and supported across relational and societal contexts.

This relational lens is echoed in the revised *Dual Process Model* of coping with bereavement *(DPM-R),* proposed by Stroebe and Schut [15], which bridges individual-level and relational models of coping. The original model *(DPM)* [16] conceptualizes coping as an oscillation between loss-oriented processes, such as confronting the reality of the death and engaging in *grief work*, and restoration-oriented processes involving adaptation, role adjustment, and reestablishing everyday continuity [16,17]. In its revised form [15], the DPM-R incorporates both intra- and interpersonal dimensions within the family context. Stroebe and Schut emphasize that coping unfolds not only within individuals but also through the relational dynamics of the family. Coping, in this framework, is not merely "the sum" [15] of individual responses, but an emergent, interdependent process.

This shift toward relational understandings of bereavement is further advanced by Breen and colleagues [12], whose work is situated at the intersection of psychosocial bereavement research and public health, reflecting a transdisciplinary approach to grief in the family context. They argue that grief is "a family affair," offering a research- and practice-oriented critique of individualistic paradigms in bereavement theory and research. The authors advocate for centering families in bereavement research and call for a conceptual integration of grief theory with family systems theory to better account for the relational and systemic nature of loss. Importantly, they argue that this perspective should inform not only research but also formal bereavement care, suggesting that family-oriented approaches may support adaptation and reduce adverse outcomes by addressing both individual and collective needs.

### Social support and family bereavement

Although not conceptually foregrounded in these frameworks, social support is implicitly embedded in the relational dynamics they describe. Existing literature widely recognizes social support as a protective factor in bereavement, associated with reduced grief-related distress and improved emotional adjustment [18,19]. Conceptually, social support is a multifaceted phenomenon embedded in and exchanged through interpersonal relationships across families, partnerships, friendships, and communities [20,21]. Shumaker and Brownell [20] broadly define it as a set of resources that serve both health-sustaining and health-compensating (i.e., stress-related) functions. Such support may foster affiliation, connection, and a sense of security [20] and is commonly classified into emotional, instrumental, informational, and appraisal dimensions [22].

Empirical research suggests that social support in bereavement may operate through both a main effect—enhancing overall well-being [23]—and a stress-buffering effect—mitigating the psychological impact of grief-related stressors [19]. While professional or community-based support may be available, bereaved individuals

most often rely on informal networks, particularly family, for emotional and practical support [24]. However, the extent to which this support is experienced as helpful varies widely and depends on how well it aligns with the bereaved person's needs—particularly in terms of its timing, amount, function, and the relationships involved [25]. Emotional support, characterized by compassion and presence, is often perceived as most helpful [18]. When such support is absent or misaligned with the bereaved person's needs, it can intensify grief-related distress [25]. Moreover, family relationships and dynamics themselves are affected by the loss, and support may be complicated by differing grief responses [15] or by interpersonal strain, which is often reported in the context of traumatic grief, such as suicide loss [18]. Thus, the family represents not only a key context in which grief is experienced, but also an important—yet complex and sometimes vulnerable—source of social support, especially in suicide bereavement, which is marked by distinct characteristics that can further challenge family relationships and the provision of support.

## Impact of suicide loss on families

Suicide bereavement is marked by unique emotional and relational challenges that differentiate it from other forms of loss. Jordan [26] identifies three dimensions in which these differences are most apparent: the thematic content of the grief, the social dynamics surrounding the bereaved, and the impact on family systems. SLS often face significant challenges in making sense of the circumstances surrounding the death, a process that can be accompanied by intense feelings of guilt, self-blame, anger, and abandonment [26]. These reactions are often compounded by the stigma that continues to surround suicide, which manifests in public perceptions of bereaved families as neglectful, dysfunctional, or responsible for the death [27,28]. SLS frequently perceive their communities as avoidant or judgmental, contributing to feelings of rejection, isolation, and social exclusion [29–31]. This public stigma is often internalized by SLS as self-stigma, reinforcing shame and social withdrawal in suicide bereavement [28,32]. Together, public and self-stigma pose significant barriers to self-disclosure, help-seeking, and access to social support— both within families and in the broader community [28,32,33]. As Jordan [26] notes, these challenges are often compounded by the emotional toll and mental health struggles of family members, along with communication barriers that can further strain familial relationships and hinder adaptive coping.

The third dimension, the impact on the family unit, remains relatively underexplored in the empirical literature. Jordan and colleagues [34] provided one of the earlier efforts to conceptualize how sudden and traumatic deaths can disrupt intra-family processes, highlighting patterns such as impaired communication and emotional withdrawal, especially in families where grief is not acknowledged or openly discussed. Their work, while based primarily on clinical observation and preliminary research, contributed foundational insights into systemic family responses to such losses. Building on this, a literature review by Cerel et al. [35] identified patterns of blame, secrecy, and emotional withdrawal that can emerge following a suicide, straining communication within bereaved families and social networks. More recent qualitative work by Creuzé et al. [36] expands this understanding, with participants describing suicide loss as "an extremely violent and traumatic event that shakes the family to the core." Their findings revealed dynamics of taboo, inhibited emotional expression, and interpersonal conflict that can disrupt communication and erode family cohesion.

These experiences have important implications for both social support provision and mental health outcomes. Research indicates that SLS frequently report unmet support needs [37–39], which have been associated with greater psychosocial distress and an increased risk of suicidality [38]. Inadequate social support can deprive individuals of essential coping resources [18] and is linked to heightened psychosocial distress and mental health risks among SLS, including symptoms of suicidality and depression [38,40]. Levi-Belz and colleagues found that interpersonal factors such as thwarted belongingness are predictive of heightened suicidal ideation and complicated grief [41,42]. Conversely, secure attachment, through mechanisms such as self-disclosure and perceived social support, can foster posttraumatic growth among SLS [43], underscoring the value of social connectedness and relational security.

### Research gaps and rationale

While research has shown that suicide loss profoundly impacts emotional and relational processes within families, there remains limited understanding of how family support is experienced. To date, no studies have explored in depth how family members support one another following the suicide of a family member. Existing literature has largely focused on individual psychological outcomes associated with perceived social support and on general support needs, with relatively little attention to the relational mechanisms that facilitate or hinder support within familial systems. This gap underscores the need for qualitative research that examines the lived experience of family support in suicide bereavement.

### Study aim and contribution

The present study addresses these gaps by exploring how individuals bereaved by the suicide of a family member experience support within their families. Specifically, it investigates the characteristics and determinants of perceived family support.

In conducting this study, we did not impose an a priori definition of 'family.' Instead, we allowed participants to define for themselves what family meant in the context of their bereavement. This approach aligns with Breen and colleagues' [12] conceptualization of family as "a web of relationships [...] that is individually experienced; may have legal, biological, and/or relational bases; and exists within social, temporal, and cultural contexts." Such a perspective recognizes that family is not a uniform construct, but a system shaped by personal meaning and sociocultural context. Nevertheless, to ensure analytic focus on intra-family support dynamics, we applied an operational boundary: participants included in this analysis had experienced the suicide of someone with whom they shared a familial context—understood as a connection that they themselves identified as part of their family system.

This exploration contributes new insights to inform postvention strategies that strengthen familial support systems and foster both personal and relational coping after suicide loss.

## Materials and methods

### Ethics statement

The study was approved by the Ethics Commission at Ulm University (application number: 374/18) and conducted in accordance with the Declaration of Helsinki. Participants were informed about the study's aims, procedures, and potential risks during a preliminary educational phone call and via written materials. All participants provided written informed consent prior to participation and data processing. Participants also received information on available mental health and bereavement support services.

### Study design and setting

This study is part of *DE-LOSS*, a mixed-methods project conducted in Germany between 2021 and 2024 that investigates experiences and determinants of social support following suicide loss. As part of this project, we employed a qualitative interview design to explore subjective perceptions, meanings, and the interpersonal and social dynamics of social support following suicide loss. The present analysis focuses specifically on how SLS experience support within their families following the suicide of a shared family member. To ensure methodological transparency and rigor, we adhered to the *Consolidated Criteria for Reporting Qualitative Research (COREQ)* [44].

### Recruitment and sample selection

The recruitment period for this study was from 17/01/2022–31/03/2022. Participants were recruited via multiple channels, including newspaper advertisements in southern Germany and outreach to national SLS support groups. These advertisements outlined the study's purpose, procedures, compensation, and contact details. A total of 106 individuals expressed

interest by contacting the research team via telephone, email, or an online form. Interested individuals then participated in telephone screenings conducted by an experienced qualitative researcher (FM) to confirm eligibility and clarify the study process. These conversations also served to address any concerns and ensure informed participation.

Eligibility criteria were as follows: (1) age 18 or older, to ensure legal capacity to consent; (2) fluency in German, to enable in-depth qualitative interviewing and ensure data quality by supporting participant comfort and nuanced self-expression; (3) the suicide loss of someone significant to them, defined as 'someone important to you and your life,' to allow for inclusive and participant-driven definitions of relational closeness; (4) age 14 or older at the time of the loss, as the study focused on adult and late adolescent experiences of social support; and (5) a minimum self-rated emotional distress score of 3 on a 5-point scale (1 = not at all; 5 = extremely), to include individuals who perceived themselves as emotionally impacted by the loss, affected by grief, and potentially in need of support.

A purposive sampling strategy was initially used to ensure diversity across participant characteristics such as gender, age, relationship to the deceased, and use of formal support services. We aimed for a sample size of 20 participants, which was deemed methodologically appropriate and practically feasible, based on prior experience with qualitative research on emotionally complex and deeply personal topics. Participants who did not take part in an interview were pre-registered for a follow-up online survey conducted within the broader DE-LOSS project.

Of the 20 interviews conducted, five were excluded from this analysis. Based on the study's focus on intra-family support dynamics, we included only participants who had experienced the suicide of someone with whom they shared a familial context. Three interviews were excluded on this basis, as they involved losses of a partner or close friend whose death was not experienced as part of a shared family dynamic—for example, a 23-year-old participant who had lost his partner, with whom his family had minimal contact. Additionally, two interviews involving assisted suicide were excluded due to the distinct contextual considerations associated with those cases [45]. The excluded interviews informed related publications within the DE-LOSS project, addressing non-familial social support [29] and the experiences of individuals involved in assisted suicide [46]. These exclusions were made during data analysis, as the specific thematic focus on intra-family support emerged inductively rather than being predetermined. Accordingly, the final analytic sample of 15 participants represents a thematically relevant subsample of the purposively recruited group. No additional interviews were conducted, because data collection for the qualitative sub-study had already concluded. The final sample size remains in line with prior qualitative studies on suicide bereavement (e.g., [36]) and was sufficient to explore a diverse range of perspectives within the study's defined scope.

## Sample

The final sample comprised 15 participants (self-reported gender: 9 female, 6 male) who had experienced the suicide of a family member. All participants described the bereavement context analyzed here as embedded within a family of origin and/or legally recognized kinship system. Participants ranged in age from 23 to 64 years (M = 47.13, Md = 48), and their average age at the time of loss was 39.35 years (range: 14–63; Md = 42). The deceased comprised four parents, three siblings, five children, two spouses, and one grandparent. One participant reported multiple familial suicide losses (uncle, godfather, and father). The mean age of the deceased was 44.12 years (range: 15–83; Md = 43). Table 1 provides a summary of participant demographics and loss-related characteristics.

## Data collection

All participants completed a brief demographic questionnaire assessing age, gender, religious belief, migration background, education, employment, living situation, and suicide loss characteristics. Interview data were collected using a semi-structured, problem-centered interview format [47]. This approach ensured alignment with the study's core themes while allowing flexibility to explore personally meaningful or unforeseen topics. The interview guide was developed through a comprehensive literature review on experiences of suicide bereavement and social support theory, and refined

**Table 1. Participant demographics and loss-related characteristics.**

| Participant code | Gender | Age | Age at loss | Years since loss | Deceased person | Deceased's age |
|---|---|---|---|---|---|---|
| P01 | Female | 38 | 14 | 25 | Grandfather | 83 |
| P02 | Female | 37 | 19, 22, 36 | 19, 16, 1 | Uncle, godfather, father | 42, 43, 70 |
| P03 | Female | 23 | 20 | 3 | Brother | 23 |
| P04 | Male | 39 | 22 | 17 | Brother | 28 |
| P05 | Female | 33 | 31 | 2 | Father | 66 |
| P06 | Female | 35 | 32 | 3 | Mother | 51 |
| P07 | Female | 55 | 42 | 13 | Father | 77 |
| P08 | Female | 45 | 45 | 1 | Husband | 51 |
| P09 | Female | 48 | 47 | 2 | Son | 16 |
| P10 | Female | 54 | 51 | 3 | Son | 22 |
| P11 | Male | 60 | 54 | 7 | Daughter | 15 |
| P12 | Male | 57 | 56 | 1 | Brother | 53 |
| P13 | Male | 60 | 57 | 3 | Son | 23 |
| P14 | Male | 59 | 58 | 1 | Son | 26 |
| P15 | Male | 64 | 63 | 1 | Wife | 61 |

in consultation with academic peers and a participatory advisory board of SLS and postvention experts. This collaborative process ensured both scientific validity and relevance to lived experience.

The guide covered themes such as perceived social support and responses from both close and extended social networks, as well as unmet support needs. Each theme was introduced with narrative-generating prompts (e.g., "Can you tell me about..."), followed by optional follow-up questions to enhance depth and nuance. An overview of the interview guide, including theme summaries and narrative prompts, is provided in S1 Table (found in the Supporting information).

Interviews were conducted between 01/03/2022 and 04/05/2022 via secure online video conferencing due to COVID-19 restrictions. Written informed consent was obtained in advance and reconfirmed verbally before starting the audio recording. Interviews lasted between 48 and 93 minutes and were audio-recorded in full. Participants received €30 compensation and were provided with information about relevant psychosocial support services. Brief rapport-building and debriefing sessions were conducted before and after each interview. Participants were free to skip questions, pause, or end the session at any point, though none chose to do so. At the conclusion, participants were invited to share any final thoughts and reminded that they could contact the research team with any further reflections or concerns.

## Data analysis

Audio recordings were transcribed and pseudonymized, with identifying details such as names and locations removed. Data were analyzed using Kuckartz and Rädiker's [48] *Qualitative Content Analysis*, a systematic, multi-step approach that combines deductive and inductive coding. The coding system was developed collaboratively through a discursive-communicative process to ensure coding quality and validate interpretation. All analyses were conducted using MAXQDA 2022, a software tool for qualitative data analysis.

First, the research team (FM, NO) conducted an initial exploration of the data, reading transcripts to familiarize themselves with the material and reflect on potential biases. FM documented reflexive memos throughout the study to examine assumptions, preconceptions, and the influence of the interviewer's position. These reflections were discussed within the research team and the participatory advisory board to enhance analytic transparency and strengthen the interpersonal validity of interpretations.

FM compiled individual case summaries, noting key themes for further analysis. An initial coding system was developed by FM based on the interview guide and thematic observations, with clearly defined category and code descriptions. The coding system was structured hierarchically, with each category comprising multiple codes, which were later refined into subcodes. NO reviewed the system for clarity and accuracy. FM then applied the initial coding system across all transcripts. To ensure consistency, NO conducted structured reliability checks by reviewing multiple segments within each code. The number of segments reviewed varied according to the volume of material assigned to each code, ensuring representation across both high-frequency and low-frequency codes.

In the second phase, the team conducted cross-case analysis, applying primarily inductive subcoding within each main category. FM and NO independently refined segments of the data and jointly reviewed emerging subcodes to resolve discrepancies and ensure shared interpretation. FM further consolidated the coding structure by merging or clarifying overlapping subcodes, using concept maps to visualize thematic relationships. All refinements were documented in code memos, including definitions and illustrative text excerpts. The finalized coding system was reviewed in a qualitative research workshop with academic colleagues for additional feedback. No further changes were required. FM then applied the final system across the full dataset.

The final phase involved in-depth analysis of key categories, guided by analytic questions derived from the study's central research aims. FM compiled and interpreted text passages from each category, and emerging findings were reviewed in research team meetings and validation workshops with colleagues and the participatory advisory board. These discussions informed interpretation, clarified conceptual distinctions, and supported refinement of both structural and textual presentation of results.

This manuscript presents findings from one category of the coding system—*family support*—which comprises 199 coded segments and emerged as a conceptually rich domain warranting in-depth analysis. Within this category, five codes were identified and further refined into 18 subcodes. These codes were grouped into two overarching themes that served to organize the material conceptually (see Table 2). In the following sections, each theme is presented with illustrative quotes drawn from participant interviews. All participant quotes are identified by their assigned code,

**Table 2. Coding system for the category of *family support*.**

| Theme | Code | Subcode |
|---|---|---|
| Contextual factors | Grief reactions and coping patterns | Facade of strength and normalcy<br>Avoidance and withdrawal<br>Disengagement<br>Secrecy |
| | Shifts in family dynamics and relationships | Strengthened bonds<br>Reduced contact and estrangement |
| | Support roles and responsibilities | Balanced mutual support<br>Division of support roles and resources<br>Uneven support capacities<br>Parenthood and childcare |
| Characteristics | Supportive family experiences | Mutual comfort and togetherness<br>Family conversations<br>Involvement of extended family |
| | Strained or insufficient support | Emotional distance and limited communication<br>Pre-existing family strain<br>Marginalization<br>Discrepancy between expected and received support<br>Desire for proactive support |

followed by gender, age at the time of interview, and the deceased's relationship to the participant (e.g., P01/ female, 38, grandfather).

Original quotes were translated from German to English by the research team through an iterative, collaborative process involving multiple rounds of translation, rephrasing, and refinement to preserve semantic accuracy and ensure the intended meaning was conveyed in the English context.

## Results

Table 2 outlines the coding system developed through qualitative content analysis. We identified two overarching themes within the category of *family support*, reflecting how individuals bereaved by a family member's suicide experienced social support within their families: (1) *Contextual factors of family support*, referring to conditions that shaped the perceived availability and quality of support; and (2) *Characteristics of family support*, referring to experiences of support, unmet needs, and interpersonal strain. Additional quotations illustrating each subcode, including full versions of quotations cited with omissions in the manuscript, are provided in S2 Table (found in the Supporting information).

### Contextual factors of family support

This theme explores the broader emotional and relational contexts that shaped the availability and quality of family support after a suicide loss. Participants described how family members' grief reactions, changes in relationships, and the division of support roles and resources within the family influenced whether support was perceived as helpful or insufficient.

**Grief reactions and coping patterns.** Participants emphasized that varying coping strategies within families—particularly avoidant or emotionally distant responses—often created barriers to mutual support. These patterns were attributed to factors such as emotional overwhelm, family roles, and implicit expectations around strength and composure.

**Facade of strength and normalcy:** Several participants described how family members maintained a facade of strength by avoiding emotional expression and focusing on routines. These behaviors were perceived as influenced by parental or gendered expectations, such as the need to appear composed for the sake of others. While some found this isolating, others—particularly those reflecting on their role as the child in the family—viewed these efforts to preserve normalcy as reassuring, as they helped buffer fears of parental breakdown.

*My mother took over everything. […] And yes, I was pretty much alone with it. No one could take care of me, my mother was completely absorbed in her tasks, doing everything and showing no weakness.* (P01/ female, 38, grandfather)

*The fear, as a child back then, was very present in me, that my parents, that they could have become completely consumed by that [their grief]. […] That, of course, wouldn't have been much better than how everything quickly and normally continued for the rest of us.* (P04/ male, 39, brother)

**Avoidance and withdrawal:** Participants commonly described emotional withdrawal and avoidance, such as reluctance to talk about the deceased, avoidance of memory-sharing, and not visiting the grave, as common coping strategies among family members. These behaviors were often interpreted as self-protective and responses to shock but left others, especially parents, concerned.

*The children don't talk much about it. I think they still keep it very much to themselves. Sometimes, that worries me, and I have no one to ask if this is normal.* (P08/ female, 45, husband)

*And my son completely ignores it. He has never been to the grave or anything like that. Ten years ago, I would have confidently said he's over it. But now, I wouldn't claim that anymore.* (P11/ male, 60, daughter)

In some families, diverging coping styles created emotional divides, limiting opportunities for shared grieving and deepening feelings of isolation. A few participants described assuming responsibility for helping others confront their grief, sometimes encouraging them to seek professional support.

*Because my husband and my daughter, especially my daughter, didn't want to talk about it at all, I felt like I couldn't grieve in the family context. I couldn't process my grief as a mother at home.* (P09/ female, 48, son)

*I tried to make it clear to my nephew, that he urgently needs to seek help. In my opinion, he was too tough, and we men always want to appear cool. But I told him that if he doesn't deal with it now, a switch might flip in him 20 years from now.* (P12/ male, 57, brother)

**Disengagement:** Some participants observed maladaptive behaviors such as complete withdrawal, apathy, or substance use. These responses intensified household stress and were experienced as both frustrating and concerning by other family members.

*My mother was totally helpless and beside herself. Eventually, I got really angry and said to her, 'Dear God, you can't just sit there all day staring into space. Snap out of it, come back!'* (P02/ female, 37, father, uncle, godfather)

**Secrecy:** Participants identified secrecy surrounding the cause of death as another coping pattern within families. From their perspective, efforts to conceal the suicide were often driven by shame, fear of social stigma, or religious beliefs. This secrecy became a source of tension and conflict, particularly around public acknowledgment, such as how the suicide was addressed at the funeral, further compounding emotional distance within the family.

*My in-laws found it extremely difficult to cope with the fact that their son took his own life. Because it's also an admission of some kind of failure.* (P08/ female, 45, husband)

*[M]y mother-in-law […] is very devout Muslim […]. She couldn't accept it at all. It's one of the gravest sins to take one's own life in their belief. And you could tell that she repressed it so much that it was as if she didn't know about it a few weeks later.* (P14/ male, 59, son)

**Shifts in family dynamics and relationships.** Participants described how suicide loss reconfigured family relationships in diverse ways—sometimes strengthening emotional bonds, and in other cases leading to conflict and estrangement.

**Strengthened bonds:** Several participants experienced strengthened family bonds following the loss, characterized by a deeper sense of belonging, greater openness to meaningful conversations, and more time spent together. They attributed these shifts to a renewed appreciation for family and to the unifying effect of shared grief, which fostered emotional closeness.

*So, the three of us became very intensively connected. My husband, [daughter's name], and I.* (P09/ female, 48, son)

*I believe I've never talked so much with my dad in my entire life as I did afterwards. Also, in a very intimate way. That word comes to mind. I had never seen my dad cry before, and that changed something, also with my older brother. The three of us talked a lot, and sometimes just me alone with my dad, really for hours.* (P06/ female, 35, mother)

**Reduced contact and estrangement:** Other participants described emotional distancing or complete estrangement from certain family members, extended relatives, or the family as a whole. These "ruptures" (P02/ female, 37, father, uncle, godfather) were linked to differing coping styles, blame, and stigmatizing attitudes toward the deceased. In some

cases, estrangement was seen as an act of self-protection; in others, it was a reaction to moral judgment or the trivialization of the death by others.

*And I miss talking to my sister today, I have to say. Because unfortunately, she completely cut off contact with all of us. But I think she saw that as her solution.* (P04/ male, 39, brother)

*My father ultimately broke ties with the extended family […]. There were comments like 'Why didn't he [the deceased] get on a motorcycle and crash into a bridge pillar?', or 'Why didn't he hang himself in the woods?', and all sorts of things like that. And my father couldn't handle that. At some point, I also said, 'No, I just don't want anything to do with these people anymore.'* (P02/ female, 37, uncle, godfather, father)

**Support roles and responsibilities.** Participants described how support roles and resources were distributed within families, ranging from mutual care to a lack of support due to limited emotional capacity and the demands of parenthood on availability and perceived responsibilities.

**Balanced mutual support:** Some families cultivated mutual care and support after the loss. Participants who experienced this reciprocity emphasized emotional closeness and a strong sense of togetherness. (This theme is further explored under *Mutual comfort and togetherness*)

**Division of roles and resources:** Support was often divided across practical and emotional domains, typically reflecting existing family roles and personal strengths. Participants—especially women—reported assuming responsibility for emotional caregiving, sometimes without feeling emotionally supported themselves.

*My brother […] took care of a lot, organized much, because he could quickly go to the apartment [of the deceased] for documents and such. And yes, I think I was more, actually, as it is perhaps obvious, the one who comforted or tried to provide emotional support. Yes. (Interviewer: Did you feel that someone was also taking care of you?) I took care of myself.* (P05/ female, 33, father)

**Uneven support capacities:** Some family members were perceived as emotionally depleted and overwhelmed, and therefore unable to provide support to others in the family. As a result, others often assumed a sense of responsibility and protective duty, leading to unequal emotional and practical burdens.

*I had to function, I had my mother who couldn't cope, I had my child, I had my household. I had to function.* (P02/ female, 37, father, uncle, godfather)

**Parenthood and childcare:** Parenthood and caregiving responsibilities played a central role in how participants experienced support. Parents described striving to maintain family stability while supporting their grieving children—often at the expense of their own emotional needs. Despite these challenges, many acknowledged that caregiving helped them cope by providing structure and a sense of purpose.

*The mother role helped me, there are always little tasks, routine activities that just continue despite everything else. And the love for my children naturally helps me tremendously. Of course, it is simultaneously a burden because one bears responsibility for them, doesn't want to miss anything and wants to be there for them. Yet, one cannot relieve them of the burden of facing this fate or inheritance in some way. And yes, it is both, it is certainly both.* (P08/ female, 45, husband)

*Having someone to care for, yes, it creates a space where one tries to grasp the joy of life again and pass it on, especially to my son, even more so for the future. And that prevents you from falling too deeply. I found that to be helpful.* (P10/ female, 54, son)

Many parents expressed self-doubt in assessing their children's emotional well-being, initiating conversations about the suicide, or recognizing signs of distress. One mother, caring for four young children, reached a point of emotional overload where support within the family was no longer sufficient. She described her struggle to access help through child welfare services, highlighting both the emotional toll and structural barriers.

*I asked if there is any support for the children, someone who […] keeps an eye on them together with me, to make sure they are doing well. This responsibility lies solely with me, and I find it really challenging. The youth welfare office told me that such support exists, but only when a child is already in trouble and showing behavioral issues […]. Preventively, it doesn't exist, and I didn't get any support. So when it comes to supporting and accompanying the children, I am left on my own. […] And if I couldn't handle it now, I could fall into depression myself, and then the children would be completely on their own. […] I would have to pathologize one of my children to possibly get help. And I don't want to subject them to that pressure, because they are doing the best they can.* (P08/ female, 45, husband)

Several participants expressed a need for family-oriented outreach, preventive care for children, and practical guidance on how to support young people through suicide bereavement. The absence of such resources was experienced not only as a systemic gap but also as a source of emotional strain within the family, leaving parents feeling isolated in their role as grieving caregivers.

### Characteristics of family support

This theme explores participants' experiences of both supportive and unsupportive interactions within their families following the suicide. While many described meaningful emotional support and mutual care, others recounted experiences of disappointment, unmet needs, or relational strain.

**Supportive family experiences.** Participants described various ways in which family members supported each other after the loss. Emotional presence, shared routines, meaningful conversations, and help with childcare emerged as central to feeling supported.

**Mutual comfort and togetherness:** Participants emphasized the importance of mutual comfort and togetherness in providing emotional support within the family. A strong emotional bond was seen as essential for coping with grief, often expressed through physical presence, nonverbal gestures, and shared routines. This sense of togetherness transformed individual grief into a shared experience, easing feelings of isolation and emotional overwhelm. Daily activities, such as shared meals and taking walks together, offered comfort, belonging, and continuity. In these moments, the distinction between providing and receiving support shifted toward simultaneous reciprocity, fostering companionship and solidarity following the loss.

*I believe the family really carried one another through it, incredibly so.* (P11/ male, 60, daughter)

*We always had breakfast together, and dinner too. Even today, I still regularly go to my daughter's for meals.* (P15/ male, 64, wife)

*I was just totally sad and had to cry, so I withdrew because I didn't want to do it openly in front of the children. […] And then the girls came and sat with me, and actually really comforting me. And the oldest gave me a look that said, 'We're all in this together.' That was a really good feeling.* (P08/ female, 45, husband)

Support was also described in terms of emotional sensitivity. Participants valued family members who respected their individual grieving styles. A quiet presence—free from pressure or expectations to talk—was perceived as compassionate and attuned to their emotional needs.

*But it was often very silent, actually. I mean, yes, [husband's name] noticed that it affected me a lot. But, of course, I didn't want to talk about it every evening. It's just that, to put it bluntly, things are just terrible. He probably just gave me the space or the peace to grieve without saying much.* (P06/ female, 35, mother)

In addition to acknowledging individual grief trajectories, participants emphasized that maintaining a supportive family environment required mutual understanding and the absence of "blame games" (P11/ male, 60, daughter).

*What was really good is that, from the beginning, there was never any blame assigned in any direction. I find that very important because I've seen marriages fall apart because of that.* (P11/ male, 60, daughter)

**Family conversations:** Open communication emerged as a vital form of support. Talking about the deceased, sharing emotions, and reflecting together on the circumstances surrounding the suicide helped participants process and accept the loss. Even brief, spontaneous acknowledgments from family members were described as supportive and meaningful.

*There's my youngest brother, whom I see often. With him, with his wife, I can really talk well about it, also about my wife. I can also cry there.* (P15/ male, 64, wife)

*What also made me happy, recently, my older daughter, who is 13, came to me. We were sitting together, and then she said: 'Mom, you're doing a good job.' That moved me a lot.* (P08/ female, 45, husband)

Ongoing family conversations played a crucial role in helping participants gradually come to terms with the suicide and understand its context. These dialogues supported shared meaning-making, enabling families to process the loss together, address lingering questions and construct a coherent narrative. In some cases, this took the form of 'informal investigations'—revisiting memories, analyzing patterns, or reading the deceased's diaries—to better understand the events leading up to the suicide. For many, this process led to an understanding of the suicide as the result of an underlying mental illness, fostering a more compassionate and destigmatizing perspective.

*The conversations with my dad and my brother. So, these long, intense conversations. That really helped me a lot to process it.* (P06/ female, 35, mother)

*But, actually, I only talked about it with my sister, because she, of course, was the one who could best assess what it was about or what had triggered it. And she, of course, has also recognized and known all her life, obviously, how we grew up.* (P04/ male, 39, brother)

*We all pored over her diaries […], and eventually we said: 'Actually, we should have known, but we didn't.' […] It was simply a disease that we didn't recognize. It could have been cancer or something else.* (P11/ male, 60, daughter)

These conversations helped participants process overwhelming feelings of guilt and responsibility. Through shared reflection, many were able to contextualize their perceived failures and recognize the limits of what could have been known or prevented. In opening up to one another, family members found validation and emotional relief, easing the burden of self-blame.

*The conversations with my dad and my brother helped me a lot, also to share this burden a bit. So, this guilt or-, yes, these feelings of guilt, that I should have done something. When of course, on the other hand, my dad was in the same situation. Yes, as a partner he could have supported her more, or more-. These are the thoughts one has. When I think about it rationally again, I think: You can't change anything. The disease can simply take such a course. I can't significantly influence it by behaving differently now. But you're not thinking rationally at that moment. And that's why it just*

*helps if someone takes some of the guilt away from you, the weight off your shoulders. Because you just feel guilty.* (P06/ female, 35, mother)

**Involvement of extended family:** Extended family members played an especially important role in the immediate aftermath of the suicide. Participants described their presence as comforting in a way that felt natural, offering a sense of stability and continuity during a period marked by shock and disorientation.

*So, amidst all the commotion here in the house in the evening, [husband's name]'s older brother and his wife came over. And the niece came over too. They had, of course, seen the flashing lights and everything in front of our house. And the younger brother, too. They were there as well. They stayed until two or three in the morning when we decided to try to sleep. And there was always someone there every day, offering support and being there with us. They endured it with us. They were really, not intrusive, but very loving. They kept asking how we were doing. And if they could help, whether we wanted to talk or not.* (P09/ female, 48, son)

*My sister was also there right afterward, and I have to say it was a big help and a big effort on her part because she also has three children. And she was there, she was definitely there for about two weeks.* (P08/ female, 45, husband)

Relatives also helped with practical tasks, including planning the funeral, clearing out the deceased's home, managing documents, and handling household chores. They also played an active role in caring for bereaved children. This involvement not only provided comfort to the children but also relieved some of the daily burdens that grieving parents faced, allowing them space for personal grieving and moments of emotional release.

*I could also say to my sister, please, right at the beginning, please look after [child's name] specifically, and I'd be grateful if you occasionally took him out on a Sunday for an outing, something nice, so that my husband and I, as a couple, can walk and talk and be sad as well. But he should also have good experiences. So, it's not like we want to hide everything or anything like that. But he shouldn't be overwhelmed.* (P10/ female, 54, son)

**Strained or insufficient support.** Several participants described relational distance, unmet needs, or emotional neglect within their families. These experiences often amplified their distress and feelings of isolation.

**Emotional distance and limited communication:** Some participants—especially those who experienced their loss during adolescence or young adulthood—described longstanding family dynamics marked by emotional distance and limited communication. These patterns persisted after the loss, leaving participants feeling unsupported and unable to seek comfort from their parents.

*In our family, warmth was simply not present. […] So, emotions were never to be expressed or allowed. That's why I wouldn't have felt comfortable in those days, not even to cry or talk about it with my parents.* (P04/ male, 39, brother)

**Pre-existing family strain:** Pre-existing family challenges, such as parental substance use, conflict, or neglect, compounded the psychological impact of the loss. One participant, who lost her grandfather during adolescence, described intense guilt and psychological distress, including self-harming behavior. She experienced her attempts to communicate distress to her parents as being met with dismissiveness, deepening her sense of abandonment. She interpreted this response as part of a broader family pattern of emotional unavailability and avoidance.

*And for me, it felt like-. Now I'm to blame for what my grandpa did. I'm to blame for what my grandpa did to my mother, how it affected her, making it [mother's alcohol use] worse. It was like a spiral of guilt. And my self-harm was mentioned*

*once: 'Don't do that anymore, it's not good.' But there was no suggestion of psychological help, or even just some advice, nothing at all.* (P01/ female, 38, grandfather)

**Marginalization:** Some participants felt overlooked or excluded within their families, often in ways shaped by generational hierarchies or implicit assumptions about emotional maturity. One participant, aged 20 at the time of her brother's death, for example, felt infantilized—spoken *about* rather than spoken *to*—and experienced a lack of direct engagement with her grief.

*[B]ut sometimes I feel that in my extended family, which isn't very big, but includes aunts, grandma, and so on, that-. I'm still seen as the child in the family. And I'm not asked how I'm doing. My mother is asked how I am, and my father is asked how I am, but not me. I'm still treated like the child who doesn't understand. […] And to some extent, [I'm] maybe even overlooked.* (P03/ female, 23, brother)

**Discrepancy between expected and received support:** Several participants described a disconnection between the support they hoped for and the support they received from both immediate and extended family members. Offers of support were often perceived as passive, vague, or conditional—placing the burden on the bereaved to initiate contact or ask for care. This dynamic contributed to disappointment and emotional distance.

*Yes, because my other siblings called at one point and said to me, '[Participant's name], if you're not doing well, you can come to me.' And that was it.* (P15/ male, 64, wife)

Some participants recalled pivotal moments of acute need, such as the night of the death, when support from close family members was absent. These moments were experienced as especially painful and contributed to growing emotional distance and relational strain.

**Desire for proactive support:** Across these accounts, participants expressed a need for proactive and sustained engagement. Many longed for small gestures that convey presence and acknowledgment—without requiring them to articulate their needs.

*You don't always have to talk about suicide. […] they could still say, 'Come on, let's go for a walk for an hour, or meet somewhere, anywhere, and just walk and talk a bit.'* (P15/ male, 64, wife)

## Discussion

This study examined how individuals bereaved by a family member's suicide experience support within their families. Family support emerged as a dynamic, relational process shaped by multiple interacting factors, including coping styles, communication patterns, caregiving roles, and the specific challenges of suicide loss, such as stigma and guilt. These findings highlight the interdependent nature of familial support—intersecting with both individual and collective coping—and the conditions that foster or inhibit it. In the following sections, we further examine these dynamics and discuss their implications for facilitating social support within families bereaved by suicide.

### Facilitators of supportive family processes

Family cohesion emerged as a crucial factor in sustaining support, fostering stability and shared grieving. Participants described physical closeness and maintaining everyday routines as ways to promote emotional connection and mutual responsiveness. These findings align with studies conducted in France by Creuzé et al. [36] and in the United Kingdom by Pitman et al. [38], which highlight how "the family itself" [36] can serve as both an active and latent source of support—the

latter providing reassurance that support remains available even when not actively sought. Building on Levi-Belz's [49] work in Israel on perceived belonging and posttraumatic growth, our findings suggest that family cohesion fosters inclusive environments that facilitate equitable access to emotional support within families.

Open discussions about guilt and responsibility helped SLS alleviate these emotions and foster cohesion through shared experiences. This aligns with Shumaker and Brownell's [20] conceptualization of social support, which emphasizes self-disclosure and active listening as key relational processes in creating a supportive environment. Eisma et al. [50] similarly describe emotional expression as an "adaptive counterpart" to suppression, facilitating cognitive processing in bereavement. Furthermore, research has identified self-disclosure in suicide bereavement as a protective factor against complicated grief [42,51] and a facilitator of personal and posttraumatic growth [43,52], referring to positive psychological changes following the loss.

Another key finding was the importance of shared reflections on the circumstances of the death in helping families develop a more self-compassionate and destigmatized understanding of the suicide. In line with research by Jordan [2], participants described how discussing the deceased's mental health history and personal challenges helped reframe the suicide as a complex outcome, rather than a failure of the family to prevent it. This finding reflects the broader role of meaning reconstruction in grief, aligning with the constructivist grief framework proposed by Gillies and Neimeyer [53], which identifies sense-making as a crucial coping mechanism in traumatic losses. Prior studies have shown that sense-making enables SLS to construct a narrative that is both personally "bearable" [2] and "socially acceptable" [54], thereby reducing suicide-related stigma. Castelli-Dransart [54], in a Swiss study, found that SLS who reject reductive or moralizing interpretations of suicide often come to understand their loss in ways that support personal growth and reengagement with life. Our findings echo this process, as participants who engaged in collective sense-making by considering systemic explanations reported not only gaining acceptance and relief from self-blame but also experiencing improved emotional connection within the family. Supporting this, U.S.-based studies by Currier et al. [55] and Rozalski et al. [56] identify sense-making as a key protective factor in bereavement after violent deaths, including suicide, mediating the impact of such losses on complicated grief symptomatology.

## Barriers and gaps in family support

A central challenge to mutual family support was differences in grieving and coping styles. Participants who preferred to express their grief verbally often felt emotionally distant from family members who adopted more inward-oriented coping or avoidant strategies. The absence of shared or complementary coping styles within families frequently led to a perceived lack of support. These dynamics may reflect a normative bias toward verbal expressions of grief, as participants often perceived inward-oriented coping as problematic, and more reserved family members as avoidant. This finding echoes Gilbert's [13] concept of *differential grief*, which emphasizes that differences in grieving styles can lead to misunderstanding and strain within families. Similarly, Breen and O'Connor [57], in an Australian study, found that disparities in emotional expression and memory-sharing may contribute to familial tension and estrangement following sudden loss.

Parental coping strategies significantly shaped family support dynamics. Parents often adopted *protective buffering* [58], a coping strategy in which individuals conceal their emotions in order to shield others from distress. This approach inhibited open dialogue and limited mutual support. Parents frequently described the tension of grieving their own loss while maintaining emotional availability and protection for their children, often feeling unequipped to assess and meet their children's needs. As one mother shared, without external support, tending to her own needs while caring for her children felt nearly impossible. These findings align with Wray et al.'s [59] systematic review, which highlights that while caregiving can provide meaning, it often limits parents' capacity to process their own grief. Paradoxically, protective buffering may ultimately increase distress for both the individual employing this strategy and those they seek to protect [15,59,60]. Alvis et al. [61] found that avoidant parental coping correlates with maladaptive grief in bereaved children and adolescents,

suggesting that such behaviors may be modeled and internalized through social learning. This risk may be particularly pronounced in suicide bereavement, where stigma [30] and cultural taboos [62] further discourage open discussions about the loss.

Family support was also shaped by pre-existing relational dynamics. Some participants described how emotional distance and communication barriers persisted or intensified after the loss, aligning with Hayslip and Page's [14] family systems approach, which emphasizes that pre-existing relational patterns persist in family grief dynamics. Research suggests that family communication styles, such as conversation-orientation versus conformity-orientation [63], play an important role in how families adapt to the death of a family member [14]. In our study, adolescents and young adults (aged 14–21 at the time of their loss) were especially affected by parental distance, which contributed to feelings of abandonment and self-blame. These findings echo Alvis et al.'s work [61,64], which highlights how parental grief expression and communication styles shape key caregiving "domains," such as protection, reciprocity, and guided learning, that influence adolescent bereavement outcomes [64]. Without developmentally appropriate bereavement support, young people may become particularly susceptible to shame, maladaptive coping, and adverse mental health outcomes [65,66]. This risk may be amplified in suicide bereavement, where grief is compounded by "circumstance-related distress" [66].

Tensions around suicide disclosure further restricted mutual support. Participants who sought open acknowledgment of the suicide—within or beyond the family—expressed frustration with family members who avoided or appeared to deny the suicide. These dynamics contributed to relational strain and emotional distancing. Similarly, Azorina et al., in a UK-based study, describe how suicide can become "the elephant in the room" [67]—an unspoken subject that nonetheless shapes family interactions. Conflicting perspectives on disclosure underscore how the context of a loss—including its personal impact and the meanings attributed to it—shapes family bereavement [14]. As reflected in participants' accounts and supported by previous research, non-disclosure may be rooted in suicide-related stigma, including shame, fear of judgment, and cultural or religious beliefs [28,67–69]. Although silence and non-disclosure can serve as protective coping strategies, they may also compound isolation and hinder access to social support networks [30,32].

### The role of extended family

Our study highlights the critical role of extended family members in providing both emotional and practical support, particularly in the immediate aftermath of suicide loss. One key mechanism that emerged is what we term the *proxy support effect*, whereby extended family members' support for children also indirectly supports parents. By assuming caregiving responsibilities and engaging children in shared activities, relatives provided parents with time and emotional space for personal *grief work* [16]. This support also helped alleviate parental concerns about the emotional burden their grief might place on their children. These findings align with Walsh's family resilience framework [70], which conceptualizes resilience as a systemic, relational process rather than an individual one. In line with this framework, our study reinforces the stabilizing and supportive role of "cooperative parenting/caregiving teams" [70]—often embodied by extended family members.

While extended family networks can provide a valuable combination of practical assistance and emotional support in the intimate home environment [59,70], some participants reported unmet expectations or disappointment with the level of proactive engagement received from relatives. Additionally, our findings reveal stigma-related barriers within these extended family networks. In some cases, moralizing responses to the suicide or judgmental remarks about the deceased led SLS to sever ties with extended family members. This highlights that stigma in suicide bereavement extends beyond direct blame toward the bereaved to include *vicarious stigma* [71]—distress experienced when others express negative attitudes toward the deceased. When others respond with blame toward the deceased rather than empathy toward the bereaved, SLS may feel their grief is invalidated—a phenomenon known as *disenfranchised grief* [72], in which grief is not sufficiently acknowledged or validated by one's social environment.

## Limitations

While this study offers new insights into family support after suicide bereavement, several methodological and interpretive limitations merit consideration. As is typical in qualitative research, the sample was non-representative and not designed to support statistical generalization. Rather, the aim was to explore diverse, in-depth perspectives on family support in the context of suicide bereavement. Although 20 interviews were originally conducted, five were excluded from the present analysis based on an inductively developed focus on intra-family support. While the final sample of 15 participants aligns with qualitative standards and provides a broad range of insights, this post hoc narrowing may have reduced the diversity of perspectives represented in this particular analysis.

The recruitment strategy, which utilized support groups and public calls, may have engaged individuals more comfortable articulating their personal bereavement experience, potentially due to their access to both formal and informal support. In contrast, those experiencing greater distress, stigma, or social withdrawal may have been less likely to participate. As a result, the sample may underrepresent experiences marked by relational distance, conflict, or disenfranchised grief—both within and beyond the family system. In addition, the sample was geographically and demographically specific, consisting predominantly of highly educated, non-migrant participants from Germany. These factors limit the transferability of findings to other sociocultural contexts. Future research should aim to include SLS from culturally marginalized and structurally underrepresented groups to explore the diversity of familial grief and support experiences across cultures and social classes.

Because we interviewed individuals rather than family units, the study focused on personal narratives. As a result, family dynamics, such as conflicting perceptions or misunderstandings, may have remained partially obscured. For instance, while participants often described avoidance and non-disclosure in others, they rarely framed their own behaviors as avoidant, perhaps due to social desirability, limited self-awareness, or sample bias. Future studies could employ dyadic or family interviews to triangulate perspectives, offering insight into how family coping and support are interactively constructed and experienced.

This analysis drew from a broader qualitative interview study that explored social support in suicide bereavement across multiple relational and cultural domains, including family, friends, and extended networks. While this broad scope allowed a wide range of themes to emerge, it may have limited deeper analytic focus on the specific determinants and mechanisms of family support. A more narrowly focused, theory-driven approach could yield greater specificity regarding family coping and support processes. At the same time, the inductive emergence of family themes across diverse, exploratively generated narratives may underscore their centrality in suicide bereavement.

Although gender emerged as a salient factor—particularly in caregiving, emotional expression, and support provision—the study was not designed to systematically examine gender-specific experiences of family support. Future research would benefit from applying gender-sensitive frameworks to explore how roles and expectations shape suicide bereavement and support across generations and cultural contexts.

Finally, as interviews were conducted in German and translated into English for publication, some linguistic nuances may have been lost, despite the research team's bilingual fluency. This limitation applies particularly to idiomatic and regionally specific language, which may not translate directly or fully.

These limitations reflect opportunities for future inquiry—especially through research designs that are systemic, culturally inclusive, and theory-generative.

## Toward an integrative framework for family support in suicide bereavement

Situated within existing theoretical and empirical research, our findings highlight that family support in suicide bereavement is not a static resource but a dynamic process shaped by personal, relational, systemic, and sociocultural factors. At the personal level, differential grief contributed to emotional distance, whereas self-disclosure fostered connection.

Relationally, family cohesion facilitated supportive interactions, while protective buffering, emotional distance, and parental disengagement reinforced silence and isolation. Systemically, caregiving responsibilities provided stability but also imposed emotional constraints, especially for grieving parents. Extended family offered practical and emotional support, though its impact depended on attitudes toward suicide and levels of proactive engagement. At the sociocultural level, suicide stigma influenced disclosure and open communication, whereas shared sense-making emerged as a counterweight, enabling families to challenge internalized stigma.

These findings underscore the need for multilevel, integrative frameworks that situate grief and support within family systems and broader sociocultural contexts. This aligns with Breen and colleagues' [12] call to bridge family systems theory and grief models to better capture the complexity of bereavement within relational environments. Building on this, we propose advancing a conceptual approach that integrates systemic and relational grief models, social support theory, and stigma frameworks. The following section outlines theoretical approaches that may be useful to integrate or expand upon to more fully reflect the interdependent nature of support in suicide-bereaved families. Advancing theory in this area is essential for identifying support barriers, understanding bereavement determinants, and informing effective, family-centered interventions for SLS.

Grief models such as the DPM-R [15] and Hayslip and Page's family systems framework [14] underscore the interdependent nature of grief within families, acknowledging how coping styles, communication patterns, generational roles, and demographic factors influence how bereavement is experienced. Complementary theoretical approaches to social support—drawing on relational sociology [73], symbolic interactionism [74], network theory [21], and communication theory [75]—offer lenses for examining how support is constructed, interpreted, and sustained within families. These perspectives move beyond transactional models of social support to account for the social embeddedness, communicative, and subjective dimensions of support, making them especially relevant for understanding how family members interpret and respond to each other's grief after suicide loss.

Suicide bereavement is further shaped by cultural narratives and the stigma of suicide. Research shows that both public and internalized stigma contribute to psychological distress and can limit access to support [32,52,76,77]. Our findings underscore the family as a key context in which stigma is confronted and managed, shaping how grief is experienced, expressed, and supported. Corrigan, Sheehan, and colleagues [27,28] have developed the most structured conceptualization of family stigma in this context, combining social-cognitive theory with empirical research on lived experience. Building on their work, future research should refine the concept of family stigma in suicide bereavement to encompass not only sociocultural but also interpersonal and systemic dimensions. This includes exploring how stigma, silence, and shame manifest across generations, are embedded in family roles and dynamics, and are shaped by shifting societal discourses around suicide and mental health.

Qualitative and reconstructive approaches like *Constructivist Grounded Theory (CGT)* [78] offer a strong methodological foundation for advancing theory in suicide bereavement research. While primarily inductive, CGT incorporates theoretical reasoning and iterative coding, allowing researchers to develop and refine conceptual frameworks grounded in participants' lived experience. This is particularly important in the context of suicide loss, where relational and sociocultural influences are difficult to capture through standardized approaches. By exploring how SLS construct grief, support, and stigma within family and social systems, CGT helps bridge the gap between subjective experience and theoretical advancement.

A key contribution of this study is the finding that social support functions as a co-constructed, relational process within bereaved families. While often framed as a facilitator of coping [20] or a protective buffer against distress [79], our findings suggest that support is continuously shaped by individual grieving styles and interpersonal coping dynamics. Although earlier work, including that by Shumaker and Brownell [20], has acknowledged the bidirectionality between coping and social support, noting that it "interfaces with almost every coping strategy mentioned in the stress literature," its theoretical integration into grief models remains underdeveloped. The DPM-R [15] provides a useful foundation for addressing

this gap, describing an oscillation between loss- and restoration-oriented coping that reflects patterns observed in our data. While social support is implicitly embedded in the model, it is not explicitly theorized as a relevant mechanism within bereaved families. Our data illustrate that social support both shapes and is shaped by this oscillation, as family members navigate different forms of coping while also taking on varying support roles—providing, receiving, and often engaging in both simultaneously, thereby *co-constructing* support within the family. However, these interactions are not always complementary. Mismatched coping and support needs can lead to emotional disconnection or limit the perceived availability and quality of support. Stroebe and Schut acknowledge such tensions, noting that incompatibilities in coping may amplify grief-related distress [15], but the role of social support remains peripheral in their model. We argue that explicitly integrating social support into the DPM-R would enhance its conceptual and practical relevance—particularly for designing systemic, family-centered interventions that are responsive to the relational dynamics of bereavement, including but not limited to suicide loss.

## Conclusion on directions for research and practice

This study highlights both the potential and limitations of familial support in suicide bereavement, emphasizing the need for family-centered, systemic postvention strategies alongside targeted support for bereaved parents, youth, and SLS for whom family is not an accessible source of support.

A key finding is the value of collective sense-making in coping with stigma and enhancing family support. Interventions that promote meaning reconstruction may reduce self-blame and strengthen emotional support, though further research is needed to clarify the role of sense-making in mediating social support. Suicide literacy and psychoeducation can support this process by challenging self-stigmatizing beliefs and contextualizing common grief responses, such as anger, guilt, and shame, fostering self-compassion and more supportive family dynamics. Evidence suggests that improving suicide literacy reduces stigmatizing attitudes [80], a principle that could inform grief-focused interventions to address the psychosocial consequences of stigma, including reduced social support [32] and help-seeking [81].

Interventions should also address support barriers arising from differences in grieving styles and disclosure preferences. Research indicates that relational strain in bereavement often stems from perceived, rather than actual, coping differences [15], highlighting the value of combining grief literacy with communication-focused approaches to foster mutual understanding. In connection with this, disclosure-based interventions that facilitate informed choices about sharing one's story may foster more supportive family environments. The *Honest, Open, Proud (HOP)* program [82]—originally developed as a peer-led group intervention to support individuals with mental illness in making disclosure decisions and coping with stigma—has been identified as adaptable for suicide bereavement [69]. As our findings suggest, culturally and family-centered adaptations of HOP may help reduce self-stigma and ease familial tension surrounding disclosure of suicide loss. Given the impact of public stigma and anticipated negative social reactions [28,69,83], stigma-reduction efforts should not only target SLS but also challenge stigmatizing societal narratives around suicide loss.

A key implication of this study is the importance of early interventions for bereaved caregivers, especially single caregivers. Protective buffering or avoidance may unintentionally weaken family support and reinforce avoidant coping in children, whereas open dialogue can strengthen family cohesion, and support children's bereavement process [84]. Supporting caregivers in facilitating open, developmentally appropriate communication has been linked to children's improved understanding and emotional regulation after suicide loss [85], suggesting that psychoeducational resources that enhance caregiver confidence may further strengthen family support.

Supporting bereaved children has reciprocal benefits for caregivers [86]. Participants emphasized the role of extended family in childcare, highlighting the importance of complementing informal support with formal services—especially for families without access to extended care networks. Future research should explore how varying levels of extended family involvement influence caregiver and child adjustment after suicide loss. Notably, participants also identified structural gaps in bereavement support, especially the lack of services for children who do not meet diagnostic criteria for professional care.

Our findings underscore the limited support available to adolescents and young adults, particularly when parental relationships are marked by emotional distance or limited communication. Suicide-bereaved youth remain underserved by current interventions [87]. Given the limited focus on this group in existing research, and the constraints of our small sample, future studies should explore how familial relationships, parental mental health, and communication patterns shape youth bereavement. Longitudinal research is needed to clarify how familial support influences adolescent adjustment and mental health over time.

Importantly, family support is not universally available. An uncritical emphasis on family in research and practice may reinforce normative assumptions that do not reflect all survivors' realities. For those with strained, distant, or unsafe relationships, grief may be compounded by unmet expectations and the absence of familial support. Clinicians should assess pre-loss dynamics and help SLS access non-familial sources of support, such as friends, professionals, peer groups, or online communities, which can reduce isolation and provide valuable support [29,88–92]. Future research should examine how pre-loss relational distance influences bereavement outcomes, whether this effect is intensified in suicide bereavement, and how non-familial support systems can be strengthened across age groups.

In this study, all participants described the familial context of their bereavement within a family of origin and/or legally recognized kinship systems. As such, the sample did not include individuals who identified a 'chosen family' [93] as the context of their bereavement. Nonetheless, such experiences are highly relevant—particularly among LGBTQ+ communities, where estrangement from families of origin and the formation of chosen families are well-documented practices and coping strategies in the face of stigma and marginalization [93–97]. These kinship structures may play a vital role in emotional support and grief processes for many individuals. Future research should adopt inclusive sampling strategies and conceptual frameworks to better capture bereavement experiences beyond traditional kinship ties, and to understand how chosen family networks can be strengthened to mitigate isolation and disenfranchised grief.

Religion and spirituality did not emerge as prominent themes in the interviews of this study. Future research could explore how faith, religious identity, or spiritual belief systems influence family interactions, meaning-making, and support-seeking in suicide bereavement.

Finally, although this study centered on family support, participants also identified broader systemic needs, including timely professional outreach, greater visibility of services, and assistance with logistical and administrative tasks, such as communicating with authorities or arranging for professional cleaning of the place of death. These reflections emphasize that familial support does not occur in isolation and that institutional systems—across mental health, education, and social care—must be designed to enable informal support in complex and stigmatized grief contexts such as suicide bereavement.

## Supporting information

**S1 Table. Overview of interview themes and narrative prompts.**
(PDF)

**S2 Table. Coding system for the category of *family support* with full and additional exemplary quotes.**
(PDF)

## Acknowledgments

The authors extend their gratitude to the participants for sharing their experiences. Special thanks are due to Martha Wahl, Susanne Barth, Ralf Albers, and Jörg Schmidt of the DE-LOSS participatory advisory board for their contributions throughout the research process. The authors also acknowledge Julia Wöhrle for her work as a student research assistant, and the members of the Qualitative Research Workshop at the Department of Psychiatry and Psychotherapy II, Ulm University, for their valuable feedback on the analytical process.

## Author contributions

**Conceptualization:** Franziska Marek, Nathalie Oexle.

**Data curation:** Franziska Marek.

**Formal analysis:** Franziska Marek, Nathalie Oexle.

**Funding acquisition:** Nathalie Oexle.

**Investigation:** Franziska Marek.

**Methodology:** Franziska Marek, Nathalie Oexle.

**Project administration:** Franziska Marek.

**Supervision:** Nathalie Oexle.

**Validation:** Franziska Marek, Nathalie Oexle.

**Visualization:** Franziska Marek.

**Writing – original draft:** Franziska Marek.

**Writing – review & editing:** Nathalie Oexle.

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
