## [Decision Letter · Decision Letter 0]

25 Jun 2025

PONE-D-25-24683Family Support After a Family Member’s Suicide: A Qualitative ExplorationPLOS ONE

Dear Dr. Marek,

Thank you for submitting your manuscript to PLOS ONE. After careful consideration, we feel that it has merit but does not fully meet PLOS ONE’s publication criteria as it currently stands. Therefore, we invite you to submit a revised version of the manuscript that addresses the points raised during the review process.

We look forward to receiving your revised manuscript.

Kind regards,

Sanja Batić Očovaj, PhD

Academic Editor

PLOS ONE

Journal Requirements:

2. In the online submission form, you indicated that [equests may be directed to the corresponding author and will be reviewed in accordance with institutional and ethical guidelines.].

Reviewers' comments:

Reviewer's Responses to Questions

**Comments to the Author**

1. Is the manuscript technically sound, and do the data support the conclusions?

Reviewer #1: Yes

Reviewer #2: Yes

2. Has the statistical analysis been performed appropriately and rigorously? 

Reviewer #1: Yes

Reviewer #2: N/A

3. Have the authors made all data underlying the findings in their manuscript fully available?

Reviewer #1: No

Reviewer #2: Yes

4. Is the manuscript presented in an intelligible fashion and written in standard English?

Reviewer #1: Yes

Reviewer #2: Yes

5. Review Comments to the Author

Reviewer #1: PONE-D-25-24683

Family Support After a Family Member’s Suicide: A Qualitative Exploration

June 2025

Franziska Marek, Nathalie Oexle

This qualitative analysis focuses on the nature of family support available after a relative’s suicide, noting the particular issues for adolescents lacking appropriate support within the family, and warning that it cannot be assumed that some people will have any family support available. This was very well written and methodologically generally very sound, although I do have some specific comments below.

Abstract

The two overarching themes identified in the abstract do not list any sub-themes in the abstract, so the text that follows this in the abstract does not align itself clearly to each of them: (1) Contextual Factors of Family Support, and (2) Characteristics of Family Support. Editing this to be clear which were sub-themes of which theme would convey the results better.

Introduction

Important to distinguish between ‘exposure’ and ‘bereavement’ when noting “from those who know the deceased to those who experience bereavement”.

When considering the perspectives of those from different disciplines (in the introduction and discussion), it is valuable to label the discipline of named authors/papers eg “Building on this perspective, the [identify discipline they belong to] Hayslip and Page…..” to be clear about how different disciplinary contributions come together.

Methods

What was the compensation for participants?

For the purposive sampling, the authors aimed for a sample size of 20 participants, “which was deemed methodologically appropriate and practically feasible, based on prior experience with qualitative research on emotionally complex and deeply personal topics”. Criteria for purposive sampling were gender, age, relationship to the deceased, and use of formal support services. It sounds as though individuals were chosen on this basis from available volunteers and the rest were registered for the broader DE-LOSS project. However, if 5 were excluded, why not increase from the 15 eligible subjects to a sample of 20?

Were any individuals from ‘made families’ / chosen families / a family of choice? ie those identifying as LGBTQ who had been alienated by their birth families and had instead built up a network of individuals functioning as a de facto family.

NO performed random checks – how was the randomness operationalised?

There was no mention of reflexivity.

This manuscript focuses on the category of Family Support, comprising 199 coded segments, but what happened to the other categories? Otherwise it sounds a bit like salami-slicing instead of presenting a coherent whole.

Did you capture religion? This was relevant to some quotes/themes so would be relevant to have measured, or to add to limitations.

Results:

I find the labels (P01) etc not very helpful in getting a quick sense of age, gender, kinship etc for each quote.

Formatting made it hard to know what level of theme/sub-theme one was at.

Discussion:

Need to mention that some nuances may have been lost in the translation to German, and whether fluency in English of NO/FM mitigated this.

Cultural context seems important so when mentioning the findings of other authors eg Creuzé et al. [35] – it would be important to mention where these studies were set.

Would be good to add strengths ahead of the limitations. The latter needs to use the term non-representative to be specific, and acknowledge that only 15 of the target of 20 were included, as well as other issues mentioned above.

Reviewer #2: Thank you for the opportunity to read this insightful and well-executed article on a qualitative research project. I have little to contribute because I find the article to be well-composed, providing a robust background for the knowledge gap, demonstrating a strong familiarity with existing research, and presenting results that offer a significant contribution to work research in a family context. Additionally, the discussion presents implications for both practice and further research. The only minor comment I have is in the Introduction (Lines 63-64), where I believe a reference is needed for the following sentence: "Grief is not only an individual reaction to loss but a relational process embedded within social…"

6. PLOS authors have the option to publish the peer review history of their article (what does this mean? ). If published, this will include your full peer review and any attached files.

**Do you want your identity to be public for this peer review?** For information about this choice, including consent withdrawal, please see our Privacy Policy .

Reviewer #1: No

Reviewer #2: **Yes: ** Sari Kaarina Lindeman

---

## [Author Response · Author response to Decision Letter 1]

25 Aug 2025

Response to Reviewers

PLOS ONE

Manuscript ID: PONE-D-25-24683

Title: Family support after a family member’s suicide: A qualitative exploration

Authors: Franziska Marek, Nathalie Oexle

We sincerely thank the editor and reviewers for their thoughtful and constructive feedback on our manuscript. Below, we respond to each comment in turn. Editor and reviewer comments are shown in bold, followed by our response and excerpts from the revised manuscript (edits highlighted in yellow) with line numbers.

Journal Requirements

Journal Requirement 1

Author response:

We confirm that the manuscript has been revised in accordance with PLOS ONE’s formatting and file naming requirements, as outlined in the provided templates.

Journal Requirement 2

In the online submission form, you indicated that [requests may be directed to the corresponding author and will be reviewed in accordance with institutional and ethical guidelines.].

Author response:

Our study is based on qualitative data from in-depth, pseudonymized interview transcripts. Although the transcripts have been carefully pseudonymized, they contain contextual details that, in combination, could compromise participant confidentiality if shared in full. According to the terms approved by our institutional ethics committee (Ethics Commission at Ulm University), participants consented to the use of pseudonymized interview data for scientific analysis and to the publication of selected, anonymized quotes in scholarly outputs, but did not consent to the full transcripts being made publicly available. In light of these ethical restrictions, we are unable to deposit the complete datasets in a public repository. We have instead provided detailed supporting information (S2 Table) containing additional anonymized quotes and full versions of quotes cited in the manuscript, to enhance transparency and rigor. In addition to these ethical restrictions, the interview transcripts are in German, which further limits the feasibility of public data sharing.

In response to the journal’s data policy requirements, we have revised the Data availability statement in the manuscript (lines 953–968) to explicitly meet all PLOS ONE criteria. The updated statement specifies:

1. The reason for the restriction (protection of participant confidentiality),

2. The name of the responsible institution (Ulm University), and

3. A non-author, institutional contact email for data access requests, in addition to the corresponding author as a secondary contact.

Revised manuscript excerpt (Data availability statement, lines 953–968):

The datasets generated and analyzed during the current study are not publicly available due to ethical restrictions aimed at protecting participant confidentiality. Although all interview transcripts have been pseudonymized, they contain contextual details that, in combination, could potentially allow for participant identification. In accordance with the terms approved by the Ethics Commission at Ulm University (application number: 374/18), participants consented to the use of pseudonymized interview data for scientific analysis and the publication of selected, anonymized quotations in scholarly outputs, but did not consent to full public disclosure of the transcripts. A minimal dataset supporting the study’s findings is provided in S2 Table (found in the supporting information), which contains additional anonymized quotations and full versions of quotations cited in the manuscript. Access to the complete pseudonymized transcripts may be considered for qualified researchers upon reasonable request. Formal inquiries regarding data access and restrictions may be directed to the Research Secretariat of the Section Public Mental Health, Department of Psychiatry and Psychotherapy II, Ulm University (E-mail: sekretariat-psyII@uni-ulm.de). The corresponding author may be contacted as a secondary point of information.

Journal Requirement 3

Author response:

We have reviewed and revised our reference list and added several references to the revised manuscript (e.g., references 6 and 93–97) in response to reviewer comments. These additions are described in detail below.

Reviewer's Responses to Questions

1. Is the manuscript technically sound, and do the data support the conclusions?

Reviewer #1: Yes

Reviewer #2: Yes

Author response:

/

2. Has the statistical analysis been performed appropriately and rigorously?

Reviewer #1: Yes

Reviewer #2: N/A

Author response:

/

3. Have the authors made all data underlying the findings in their manuscript fully available?

Reviewer #1: No

Reviewer #2: Yes

Author response:

Our study is based on qualitative data from in-depth, pseudonymized interview transcripts. Although the transcripts have been carefully pseudonymized, they contain contextual details that, in combination, could compromise participant confidentiality if shared in full. According to the terms approved by our institutional ethics committee (Ethics Commission at Ulm University), participants consented to the use of pseudonymized interview data for scientific analysis and to the publication of selected, anonymized quotes in scholarly outputs, but did not consent to the full transcripts being made publicly available. In light of these ethical restrictions, we are unable to deposit the complete datasets in a public repository. We have instead provided detailed supporting information (S2 Table) containing additional anonymized quotes and full versions of quotes cited in the manuscript, to enhance transparency and rigor. In addition to these ethical restrictions, the interview transcripts are in German, which further limits the feasibility of public data sharing.

In response to the journal’s data policy requirements, we have revised the Data availability statement in the manuscript (lines 953–968) to explicitly meet all PLOS ONE criteria. The updated statement specifies:

1. The reason for the restriction (protection of participant confidentiality),

2. The name of the responsible institution (Ulm University), and

3. A non-author, institutional contact email for data access requests, in addition to the corresponding author as a secondary contact.

Revised manuscript excerpt (Data availability statement, lines 953–968):

The datasets generated and analyzed during the current study are not publicly available due to ethical restrictions aimed at protecting participant confidentiality. Although all interview transcripts have been pseudonymized, they contain contextual details that, in combination, could potentially allow for participant identification. In accordance with the terms approved by the Ethics Commission at Ulm University (application number: 374/18), participants consented to the use of pseudonymized interview data for scientific analysis and the publication of selected, anonymized quotations in scholarly outputs, but did not consent to full public disclosure of the transcripts. A minimal dataset supporting the study’s findings is provided in S2 Table (found in the supporting information), which contains additional anonymized quotations and full versions of quotations cited in the manuscript. Access to the complete pseudonymized transcripts may be considered for qualified researchers upon reasonable request. Formal inquiries regarding data access and restrictions may be directed to the Research Secretariat of the Section Public Mental Health, Department of Psychiatry and Psychotherapy II, Ulm University (E-mail: sekretariat-psyII@uni-ulm.de). The corresponding author may be contacted as a secondary point of information.

4. Is the manuscript presented in an intelligible fashion and written in standard English?

Reviewer #1: Yes

Reviewer #2: Yes

Author response:

/

Reviewer#1 Comments

This qualitative analysis focuses on the nature of family support available after a relative’s suicide, noting the particular issues for adolescents lacking appropriate support within the family, and warning that it cannot be assumed that some people will have any family support available. This was very well written and methodologically generally very sound, although I do have some specific comments below.

Author response:

We thank Reviewer#1 for their careful reading and thoughtful comments. We address each point below.

Comment 1: Abstract

The two overarching themes identified in the abstract do not list any sub-themes in the abstract, so the text that follows this in the abstract does not align itself clearly to each of them: (1) Contextual Factors of Family Support, and (2) Characteristics of Family Support. Editing this to be clear which were sub-themes of which theme would convey the results better.

Author response:

We revised the Abstract to explicitly link each overarching theme to its corresponding sub-themes, thereby clarifying the structure and improving alignment with the Results section of the manuscript. Specifically, the theme ‘Contextual factors of family support’ now includes grief reactions and coping patterns, shifts in family dynamics, and the distribution of support roles. The theme ‘Characteristics of family support’ now encompasses both supportive experiences—such as emotional closeness, open communication, and shared sense-making—and insufficient support, including the marginalization of grief and emotional neglect. These revisions aim to clarify the thematic structure. To comply with the journal’s 300-word abstract limit, we made several minor wording adjustments to streamline the text without altering its meaning or tone. Please see lines 25–35 of the revised manuscript for these changes.

Revised manuscript excerpt (Abstract, lines 25–35):

Two overarching themes were identified: (1) Contextual factors of family support, including grief reactions and coping patterns, shifts in family dynamics, and the distribution of support roles; and (2) Characteristics of family support, encompassing both supportive experiences—such as emotional closeness, open communication, and shared sense-making—and insufficient support—such as marginalization of grief and emotional neglect, often linked to pre-existing family strain. The availability and quality of support were influenced by protective buffering, relational withdrawal, discomfort surrounding suicide disclosure, and the reconfiguration of relationships after the loss. Extended family members played a significant role in assisting with childcare and relieving emotional burdens on grieving parents, although their support varied depending on attitudes toward suicide and levels of proactive engagement.

Comment 2: Introduction

Important to distinguish between ‘exposure’ and ‘bereavement’ when noting “from those who know the deceased to those who experience bereavement”.

Author response:

We revised the respective section of the Introduction (lines 48–51) to more clearly distinguish between exposure and bereavement, drawing on Cerel et al.’s conceptualization of the Continuum of Survivorship. The revised text now explicitly differentiates individuals who knew the deceased, those who are emotionally affected, and those who are bereaved following the loss of someone with whom they had a close attachment. This clarification strengthens the conceptual accuracy of our framing and reinforces the focus of our study on suicide-bereaved individuals. We also added a citation of Cerel et al.’s 2014 publication (line 48), which introduced the Continuum of Survivorship, to ensure appropriate attribution and conceptual clarity.

Revised manuscript excerpt (Introduction, lines 48–51):

As Cerel and colleagues [6] have emphasized, suicide loss occurs along a continuum of exposure and impact—from individuals who knew the deceased, to those who are emotionally affected, to those who are bereaved following the loss of a close attachment and experience profound psychological and relational consequences.

Comment 3: Introduction

When considering the perspectives of those from different disciplines (in the introduction and discussion), it is valuable to label the discipline of named authors/papers eg “Building on this perspective, the [identify discipline they belong to] Hayslip and Page…..” to be clear about how different disciplinary contributions come together.

Author response:

While the cited scholars primarily work within the fields of psychology, social psychology, or public health—and therefore do not represent entirely separate academic disciplines in a strict sense—they draw on distinct theoretical traditions that inform the development of our study. To clarify these distinctions, we revised the Grief as a relational process section to explicitly describe the conceptual orientation of each contribution: Gilbert adopts a social constructionist perspective on grief; Hayslip and Page draw from family psychology and systems theory to propose a family systems framework; Stroebe and Schut bridge individual and relational models in their Dual Process Model of coping with bereavement; and Breen and colleagues advance a transdisciplinary approach that integrates psychosocial bereavement research and public health. These clarifications (lines 68–70; 75–78; 84–86; 93–97) highlight the complementary nature of these frameworks and the conceptual richness they bring to the study of family grief and support.

Revised manuscript excerpt (Introduction / Grief as a relational process, lines 68–70):

Gilbert [13], adopting a social constructionist perspecti

---

## [Decision Letter · Decision Letter 1]

6 Oct 2025

Family support after a family member’s suicide: A qualitative exploration

PONE-D-25-24683R1

Dear Dr. Marek,

We’re pleased to inform you that your manuscript has been judged scientifically suitable for publication and will be formally accepted for publication once it meets all outstanding technical requirements.

Kind regards,

Sanja Batić Očovaj, PhD

Academic Editor

PLOS ONE

Additional Editor Comments (optional):

Reviewers' comments:

Reviewer's Responses to Questions

**Comments to the Author**

1. If the authors have adequately addressed your comments raised in a previous round of review and you feel that this manuscript is now acceptable for publication, you may indicate that here to bypass the “Comments to the Author” section, enter your conflict of interest statement in the “Confidential to Editor” section, and submit your "Accept" recommendation.

Reviewer #1: All comments have been addressed

2. Is the manuscript technically sound, and do the data support the conclusions?

Reviewer #1: (No Response)

3. Has the statistical analysis been performed appropriately and rigorously? 

Reviewer #1: Yes

4. Have the authors made all data underlying the findings in their manuscript fully available?

Reviewer #1: Yes

5. Is the manuscript presented in an intelligible fashion and written in standard English?

Reviewer #1: Yes

6. Review Comments to the Author

Reviewer #1: all comments have been addressed by the authors in their response

all comments have been addressed by the authors in their response

all comments have been addressed by the authors in their response

all comments have been addressed by the authors in their response

7. PLOS authors have the option to publish the peer review history of their article (what does this mean? ). If published, this will include your full peer review and any attached files.

**Do you want your identity to be public for this peer review?** For information about this choice, including consent withdrawal, please see our Privacy Policy .

Reviewer #1: No

---

## [Editor Report · Acceptance letter]

PONE-D-25-24683R1

PLOS ONE

Dear Dr. Marek,

I'm pleased to inform you that your manuscript has been deemed suitable for publication in PLOS ONE. Congratulations! Your manuscript is now being handed over to our production team.

Kind regards,

on behalf of

Dr. Sanja Batić Očovaj

Academic Editor

PLOS ONE